# Opioid-Modulated Receptor Localization and Erk1/2 Phosphorylation in Cells Coexpressing μ-Opioid and Nociceptin Receptors

**DOI:** 10.3390/ijms24021048

**Published:** 2023-01-05

**Authors:** Guan-Yu Zhuo, Ming-Chi Chen, Tzu-Yu Lin, Shih-Ting Lin, Daniel Tzu-Li Chen, Cynthia Wei-Sheng Lee

**Affiliations:** 1Institute of Translational Medicine and New Drug Development, China Medical University, Taichung 40402, Taiwan; 2Center for Drug Abuse and Addiction, China Medical University Hospital, Taichung 40447, Taiwan; 3Integrative Stem Cell Center, China Medical University Hospital, Taichung 40447, Taiwan; 4School of Chinese Medicine, China Medical University, Taichung 40402, Taiwan; 5Graduate Institute of Biomedical Sciences, China Medical University, Taichung 40402, Taiwan

**Keywords:** μ-opioid (MOP) receptor, nociceptin/orphanin FQ (NOP) receptor, two-photon microscopy, fluorescence resonance energy transfer (FRET), single-particle tracking (SPT), lipid rafts, Erk1/2 (p44/p42 MAPK)

## Abstract

We attempted to examine the alterations elicited by opioids via coexpressed μ-opioid (MOP) and nociceptin/orphanin FQ (NOP) receptors for receptor localization and Erk1/2 (p44/42 MAPK) in human embryonic kidney (HEK) 293 cells. Through two-photon microscopy, the proximity of MOP and NOP receptors was verified by fluorescence resonance energy transfer (FRET), and morphine but not buprenorphine facilitated the process of MOP-NOP heterodimerization. Single-particle tracking (SPT) further revealed that morphine or buprenorphine hindered the movement of the MOP-NOP heterodimers. After exposure to morphine or buprenorphine, receptor localization on lipid rafts was detected by immunocytochemistry, and phosphorylation of Erk1/2 was determined by immunoblotting in HEK 293 cells expressing MOP, NOP, or MOP+NOP receptors. Colocalization of MOP and NOP on lipid rafts was enhanced by morphine but not buprenorphine. Morphine stimulated the phosphorylation of Erk1/2 with a similar potency in HEK 293 cells expressing MOP and MOP+NOP receptors, but buprenorphine appeared to activate Erk1/2 solely through NOP receptors. Our results suggest that opioids can fine-tune the cellular localization of opioid receptors and phosphorylation of Erk1/2 in MOP+NOP-expressing cells.

## 1. Introduction

Opioid receptors, belonging to the superfamily of G protein-coupled receptors (GPCRs), are the major mediators of exogenously administered opioid drugs [1]. The opioid receptor family contains three conventional members, the μ-opioid (MOP) receptor, the κ-opioid (KOP) receptor, and the δ-opioid (DOP) receptor, as well as a nonopioid branch, the nociceptin/orphanin FQ (NOP) receptor [2]. Since there is a discrepancy between the existence of merely four opioid receptor genes and the considerable pharmacological documentation for additional opioid receptor phenotypes, it is speculated that alternative mRNA splicing, posttranslational modifications, tissue distribution of receptors and scaffolding with modifications, or homo- or heterodimerization of the existing MOP, KOP, DOP, and NOP proteins could contribute to the various pharmacological profiles of opioids [3].

Human MOP and NOP receptors are coexpressed in the human brain, especially in laminae I~III of the prefrontal cortex, laminae II~V of the cingulate cortex, and the caudate nucleus of the basal ganglia [4,5,6]. MOP and NOP receptors are natively coexpressed in BE(2)-C human neuroblastoma cells, in which 60-min activation of the MOP receptor desensitizes MOP and NOP receptors via activation of G protein-coupled receptor kinase [7]. Furthermore, human NOP receptors heterodimerize with MOP receptors and can cointernalize upon agonist exposure; this heterodimerization attenuates NOP receptor-mediated inhibition of N-type channels [8]. Bifunctional NOP/MOP receptor agonists provide a wider therapeutic window with fewer side effects and attenuate the reward processing effects of opioids [9]. Therefore, the coupling effects of human MOP and NOP receptors might have profound clinical implications.

Single-particle tracking (SPT) has been applied to show that MOPs can be in various mobility states under basal conditions [10,11]. It has also been used to track agonist-dependent changes in MOP mobility over time, but the mobility states were probably affected by a diverse set of downstream effectors [12]. The dynamic monomer-dimer equilibrium of MOP was described via SPT, demonstrating that the agonist DAMGO, but not morphine, induced dimer formation correlated with β-arrestin 2 binding to MOP [13]. Confocal fluorescence recovery after photobleaching (FRAP) was used to study the regulation by different MOP agonists of receptor movement within the plasma membrane of HEK 293 cells stably expressing a functional yellow fluorescent protein (YFP)-tagged MOP, revealing that the lateral mobility of MOP was increased by DAMGO and to a lesser extent by morphine [14]. Fluorescence resonance energy transfer (FRET) in HeLa cells transfected with MOP and the dopamine D_2_ receptor (D_2_R) was employed to show that these receptors heterodimerize, and live cell imaging revealed that coexpressed D_2_R slowed the internalization of MOP after activation with DAMGO [15].

Mitogen-activated protein kinase (MAPK) cascades, particularly extracellular-signal-regulated kinases (ERKs), are activated after MOP receptor activation [16] by opioid agonists [17]. ERK stimulation by opioids might play key roles in opioid tolerance and withdrawal since it is essential for internalization/desensitization of the MOP receptor in heterologous expression systems [18] and participates in synaptic plasticity [19].

Many elements of the cAMP-mediated signal transduction pathway have been demonstrated to localize within lipid rafts/caveolae. GPCRs [20,21], G protein α and βγ subunits, adenylate cyclase (AC), nitric oxide synthase, PKA, PKCα, Src tyrosine kinase, and PI3K are signaling molecules observed in such microdomains [22,23,24,25,26,27,28,29,30]. Caveolins, the architectural components of caveolae, can regulate the phosphorylation and dephosphorylation equilibrium and restrain the dynamic interactions of membrane proteins [30]. MOP receptor-mediated AC supersensitization has been demonstrated to be cell surface receptor concentration-dependent instead of agonist-dependent. Particularly, AC supersensitization requires that MOP receptors be localized within the lipid rafts of cell surface membranes [31]. Nociceptin-induced endocytosis primarily occurs via clathrin-coated pits, and the NOP receptor is primarily internalized through the endosome compartment [32]. In HEK 293 cells, the NOP receptor exists constitutively in a detergent-resistant membrane fraction that presumably comprises a mixture of rafts and caveolae [33].

Buprenorphine maintenance therapy is currently employed to treat opioid-dependent addicts. Buprenorphine is an MOP partial agonist as well as an NOP agonist and effective KOP antagonist [34,35]. Our previous study demonstrated that buprenorphine elicits different adaptive changes in AC activity in HEK 293 cells coexpressing human MOP and NOP receptors [36]. To further understand the pharmacological profiles of buprenorphine, receptor localization within the cell and activation of Erk1/2 after exposure to morphine and buprenorphine were scrutinized in our cell model.

## 2. Results

### 2.1. Fluorescence Resonance Energy Transfer (FRET)

FRET serves as a sensitive indicator of cellular processes/activities occurring at the nanoscale and has been used to identify the supramolecular organization of cell surface molecules, receptor protein interactions on the cell membrane (i.e., membrane mapping), intermolecular interactions in live cells, etc. [37,38]. The main reason for using two-photon microscopy was the usage of a near infrared laser, which provides less photodamage and photobleaching compared to experiments using visible laser light at the same power level. Thus, two-photon microscopy enables long-term observation, as required in our experiment. However, the fluorescence signals from the transfected cells containing MOP-CFP and NOP-YFP were quite low, resulting in the laser power at the sample site being approximately 25 mW in our case, which was ranked as a high-power level. Thus, special attention should be given to the balance between image quality, laser power, and photo-damage/bleaching prior to FRET measurements. With two-photon FRET imaging, cellular behaviors responding to a natural or controlled environment can be resolved noninvasively and in real time. First, it was used to study the proximity of MOP and NOP receptors under three different experimental conditions, including pre-treatment as a control, treatment with buprenorphine (post-bur), and treatment with morphine (post-mor). The results are illustrated in Figure 1A–C, showing images obtained from the MOP-CFP-channel, NOP-YFP-channel, FRET ratio counted from the above two, and a magnified view from a white square region on the respective FRET ratio image, from left to right for each condition. Notably, the control image shown in Figure 1D (obtained from the MOP-CFP-channel at the pretreatment condition) provides a representative appearance of the receptors properly expressed on the cell surface, which is the basis of determining a suitable cell status for the subsequent analysis. 

As indicated in Section 4.2, unavoidable fluorescence cross-talk between the two channels needed to be removed from the original FRET measurements. The values of *a* and *b* were measured to be 0.218 and 0.543, respectively. Then, the net FRET intensity was calibrated, and the image of the FRET ratio (in the third column) was determined with a pixelwise resolution in terms of net FRET/CFP. Through cross-talk correction for two-photon FRET, the precision and reproducibility of the measurements was improved. To appreciate the FRET and colocalization results more clearly, a magnified view showing single cells is provided in the fourth column. In Figure 1E, the histogram of the FRET ratio for each condition represented the values measured at more than 300 time points within cells, and the peak value and standard deviation are shown. Receptor proteins on cell membranes are randomly distributed due to the heterogeneity of cell membranes and the experiments relied on overexpressed MOP and NOP, which could enforce proximity by creating a situation with an artificially high density of receptors. Therefore, there was a certain probability that MOP and NOP were in close positions before dosing, resulting in the expression of a 35% FRET ratio. In addition, the peak FRET ratio of post-bup was similar to that of pre-treatment, indicating that the colocalization of MOP and NOP was not enhanced by buprenorphine. In contrast, the histogram demonstrated that colocalization was enhanced post-mor as shown by two distribution peaks, one close to the peak FRET ratios of pre-treatment and post-bup and the other with a higher value than the other peaks, indicating that part of the adjacency of the C-terminal regions and subsequent conformational changes in MOP and NOP were distinctively triggered by morphine.

### 2.2. Single-Particle Tracking (SPT)

SPT is a quantitative measure of the transport behavior of nano-objects through heterogeneous cellular environments. It is capable of providing nanoscopic information, such as the diffusion coefficient, transport velocity, and intracellular force [39]. In general, the target particles are detected and localized on a time series of images, resulting in individual trajectories with localization precision of several tens of nanometers. The trajectory can be obtained either by a high-speed camera or laser scanning microscope, depending on the time window required to observe a biological event. In our case, the diffusion coefficient of mobilized heterodimers (in pre-treatment condition) was around 0.02 μm^2^/s, which is slower than that of quantum dots (~0.2 μm^2^/s) and faster than that of polyplexes (~0.002 μm^2^/s) diffusing in cells [40,41]. We directly used the images from FRET measurements to extract information about the underlying dynamics of heterodimers on the cell membranes and then determine the pharmacological effects of the drugs on cells. Following the above descriptions, the two-photon image of NOP-YPF fluorescence with the addition of morphine is shown in Figure 2A, where various heterodimers shown as bright spots are candidates for SPT and MSD analysis. The trajectories obtained using the TrackMate plugin of ImageJ software were overlaid on the image, as shown in Figure 2B, based on the initial guess and constraints as an estimated blob diameter of 1 μm (close to the diffraction-limited size), a linking max distance of 5 μm, a gap-closing max distance of 5 μm, and a gap-closing max frame gap of 5. Although the localization precision in this study was approximately 27 nm due to the fluorescence cross-talk between the two channels and little cell autofluorescence, it was sufficient to interpret the slow dynamics of the heterodimers exposed to drugs.

With the obtained trajectories, we then analyzed the mean speed in the three conditions. As shown in Figure 2C, the fastest motion was found in the pre-treatment condition, followed by the post-bur and post-mor conditions, indicating that the motion in the post-bur and post-mor conditions was more confined than in pre-treatment and was influenced by drug inclusion, in which an increased force occurred between heterodimer and cell membrane. Notably, the mean speed of the heterodimers (~20 nm in diameter) in the pre-treatment condition was much slower than that of 1 µm diameter particles in a liquid (~1 µm/s) [42], indicating that the intracellular force persisted without drug treatment. On the other hand, the 2D MSDs were calculated according to Equation 2 and were then analyzed to extract the exponent and diffusion coefficient shown by Equation (3). For simplicity, we transformed the MSD-time lag dependency into a log-log plot, as shown in Figure 2D, which enabled the extraction of the above parameters simply using a linear fit. The slope indicated the exponent of Equation 3 used to distinguish between directed motion (1 < *α* < 2), Brownian motion (*α* = 1), and confined motion (*α* < 1) [43,44]. The results showed that heterodimers in the pre-treatment (*α* = 0.97 ± 0.02), post-bur (*α* = 0.87 ± 0.02), and post-mor (*α* = 0.81 ± 0.04) conditions all exhibited confined motion, in which the diffusion coefficients measured at the second time scale were 0.021 ± 0.006, 0.012 ± 0.003, and 0.007 ± 0.003 μm^2^/s, respectively. By comparing the post-bur with post-mor condition results, the post-mor condition had a smaller *α* and diffusion coefficient, implying that the enhancement of the FRET ratio caused by the colocalization of MOP and NOP induced by opioids hindered the motion of the heterodimers.

### 2.3. Increased Colocalization of MOP and NOP Receptors on Lipid Rafts after Morphine Treatment

We then evaluated the effects of morphine and buprenorphine on the colocalization of MOP and NOP receptors with lipid raft membrane structures using previously established HEK 293 cells stably expressing HA-tagged MOP and/or myc-tagged NOP [36]. The right panels of Figure 3A show the lipid raft structures on the membranes of HEK 293 cells under basal conditions (vehicle treatment). The distribution of MOP and NOP on lipid rafts did not appear to change significantly after morphine or buprenorphine treatment in cells expressing MOP or NOP alone (Figure 3A, higher and middle panels; Appendix A). Five independent experiments with 4–6 individual cell images in each group were then analyzed by defining the lipid rafts (red fluorescence) as the ROI. The colocalization rates of MOP and lipid rafts (green and red fluorescence) after morphine, buprenorphine, and vehicle treatment in MOP-expressing cells were 27.19 ± 2.71%, 24.45 ± 1.36%, and 27.11 ± 3.62%, respectively. The colocalization rates of NOP and lipid rafts (magenta and red fluorescence) after morphine, buprenorphine, and vehicle treatment in NOP-expressing cells were 3.06 ± 0.48%, 2.44 ± 0.73%, and 2.08 ± 0.61%, respectively (Figure 3B). Compared to the control (27.13 ± 6.53%), colocalization of MOP and NOP (green and magenta fluorescence) on lipid rafts was slightly enhanced by morphine treatment (30.95 ± 6.17%) but not by buprenorphine treatment (26.79 ± 4.34%) (Figure 3A, lower panels), yet these changes were not statistically significant (Figure 3B).

### 2.4. Coexpressing NOP with MOP Receptor Did Not Compromise the Potency of Opioids on Erk1/2 (p44/p42 MAPK) Activation

The effect of MOP-NOP receptor coexpression on opioid-induced phosphorylation of Erk1/2 (p44/p42 MAPK) was investigated using HEK 293 cells expressing MOP, NOP, or MOP+NOP receptors. Morphine stimulated the phosphorylation of Erk1/2 with a similar potency in HEK 293 cells expressing MOP alone (Figure 4A; pEC_50_ = 7.021 ± 0.639; *n* = 4) and MOP+NOP receptors (Figure 4A; pEC_50_ = 7.548 ± 0.3556; *n* = 7) but not in cells expressing only the NOP receptor (Figure 4A). Buprenorphine also induced phosphorylation with a comparable potency in cells expressing only MOP (Figure 4B; pEC_50_ = 8.076 ± 0.285; *n* = 4) and MOP+NOP receptors (Figure 4B; pEC_50_ = 8.660 ± 0.775; *n* = 5) but with a lower potency in NOP-expressing cells (Figure 4B; pEC_50_ = 7.480 ± 0.344; *n* = 4).

## 3. Discussion

Our study is the first to show how the localization of MOP and NOP receptors and the phosphorylation of Erk1/2 were modulated by opioids in HEK 293 cells coexpressing MOP and NOP receptors.

The two-photon fluorescence of MOP-CPF and NOP-YFP (Figure 1) revealed the endogenous proximity of MOP and NOP (35% FRET ratio), and a subpopulation of the receptors displayed a higher FRET ratio (61% FRET ratio in addition to 43% FRET) after morphine addition, indicating that some of the MOP and NOP became closer in the presence of morphine. This phenomenon was not observed for buprenorphine (39% FRET). Since CFP and YFP were fused to the C-termini of MOP and NOP, respectively, in our expression system, the changes in the FRET ratio might reflect intracellular conformational changes and downstream effectors being differentially recruited by morphine and buprenorphine. Our results were reminiscent of a recent report using a novel MOP fluorescent ligand, Dermorphin_ATTO488_, and the NOP fluorescent ligand, N/OFQ_ATTO594_, in HEK_MOP/NOP_ cells that elegantly demonstrated MOP-NOP probe overlap and a FRET signal indicating colocalization [45]. However, their ligand binding sites were extracellular, while our fluorescent proteins were intracellular. Therefore, some of the differences in the elicited effects of the ligands might be due to conformational changes between the N- and C-termini of the receptors upon ligand stimulation. In cultured HEK 293 cells stably transfected with MOP, morphine was weak (2-fold EC_50_ and 15% E_max_ compared with the full agonist, DAMGO), whereas buprenorphine was incompetent in recruiting β-arrestin 2 [46]. Additionally, our previous results illustrated that after acute exposure, morphine potently inhibited AC activity, whereas buprenorphine induced compromised AC inhibition in MOP+NOP-expressing cells [36], implying that acute classic Gi/Go signaling and a prolonged receptor tyrosine kinase (RTK)-like noncanonical pathway were differentially controlled by morphine and buprenorphine.

Treatment with morphine or buprenorphine hindered the movement of MOP-NOP heterodimers, as revealed by the lower mean speed and slope of fit of MSD in our SPT analysis (Figure 2). This finding corresponded with a previous report showing a decreased diffusion coefficient of MOP after 15-min treatment with morphine [13]. Since the action of the drugs of abuse is a prolonged process, our study used a longer time scale and lower values of diffusion coefficients were obtained compared to the previous study [13]. Morphine binding to the MOP could trigger a conformational change in the adjacent alpha2A-adrenergic receptor and thereby inhibit its signaling to G(i) and the downstream MAP kinase cascade [47], hinting at a similar mechanism in signal transduction whereby the MOP-NOP heterodimer mediates conformational changes propagating from NOP to MOP and causes the rapid inactivation of MOP. The importance of the delay in MOP+NOP motion might indicate multiple regulators and downstream effectors being recruited inside the cells, hence delaying the movement of MOP+NOP. It would be pivotal to identify the molecular players of opioid-elicited cellular adaptations in the future to tease out the enigma of the side effects experienced by opioid users.

Localization of MOP at lipid rafts is required for AC superactivation [31]. The recruitment and activation of Src kinase by lipid raft-located MOP during chronic agonist treatment, which results in MOP tyrosine phosphorylation, is crucial for switching MOP signals from its initial AC inhibition to subsequent AC superactivation [48]. On the other hand, NOP localizes constitutively on the lipid raft/detergent-resistant membrane [33], and lipid rafts play an important role in the antinociceptin effect of neuropeptide FF receptors [49]. Our results demonstrated that the colocalization rates of MOP, NOP, and MOP+NOP at lipid rafts were approximately 25%, 2.5%, and 30%, respectively (Figure 3). The colocalization rate of MOP+NOP and lipid rafts was 30.95 ± 6.17% in morphine-treated cells, which was higher (yet not statistically significant) than that in buprenorphine-treated cells (26.79 ± 4.34%). The same tendency with a similar magnitude was also observed for MOP-expressing cells (27.19 ± 2.71% and 24.45 ± 1.36% for morphine and buprenorphine, respectively), suggesting that the localization of MOP and MOP+NOP on the lipid rafts regulated by opioids probably occurs through the same cellular machinery associated with MOP. Because we could not exclude the possibility that the localization of receptors was affected by the levels of receptor expression and internalization, the effects elicited by morphine and buprenorphine were compared to vehicle treatment under our experimental conditions. We speculate that coexpressing NOP with MOP could change the distribution of NOP and thereby exert distinct cellular signaling by localizing NOP onto lipid rafts. The ability of the MOP agonist to acutely inhibit or chronically superactivate AC has been shown to be attenuated with membrane cholesterol removal [50]. Cholesterol not only influences MOP localization in lipid rafts but also enhances the recruitment of β-arrestins, thereby affecting agonist-induced trafficking and signaling of MOP [51] in cell models and analgesia in mice and humans [52]. Furthermore, GRIN1 (G protein-regulated inducer of neurite outgrowth 1) can tether MOP with the G protein α-subunit, thereby regulating MOP distribution, stabilizing MOP within lipid rafts, and potentiating MOP signaling in neurite outgrowth processes [53]. Hence, it would be interesting to investigate the roles of Src kinase, cholesterol, and GRIN1 in cellular responses to opioid treatment in the context of coexpressed MOP and NOP receptors.

Buprenorphine is an NOP receptor agonist and hence was able to elicit Erk1/2 phosphorylation in NOP-expressing cells, while the MOP agonist morphine exerted little if any activation effect on Erk1/2 in cells expressing only the NOP receptor (Figure 4, middle panels). Coexpressing NOP with MOP receptors did not decrease the potency of morphine on Erk1/2 phosphorylation (Figure 4A), which is surprisingly inconsistent with the previous finding that rat MOP-NOP heterodimerization impaired the potency of MOP receptor agonists [54]. As an MOP partial agonist as well as an NOP agonist, buprenorphine was almost equally potent in inducing Erk1/2 phosphorylation in MOP- and MOP+NOP-expressing cells (Figure 4B). In addition to the species difference between rats and humans, we speculate that the decreased potency of the MOP agonist might be neutralized by the effect of the NOP agonist buprenorphine, a bifunctional opioid.

Our study had several limitations. First, the heterodimerization process might differ in native cells that do not overexpress MOP and NOP receptors. In functional assays, drug efficacy can markedly differ between cell systems with overexpressed receptors compared with native systems, especially for partial agonists. Morphine is a partial agonist in native membrane systems (e.g., the thalamus) and a full agonist in cells overexpressing MOP [55]. A similar rule may also apply to beta-arrestin recruitment and Erk1/2 activation. Second, further experiments are needed to determine whether the effects observed are reversed with selective antagonists and the effect of selective MOP and NOP agonists that produce receptor internalization since the side effects of morphine might stem from the lack of receptor internalization and recycling [56]. Third, because the optical sectioning capability (i.e., submicron scale) provided either by two-photon microscopy or confocal microscopy makes it difficult to focus on the plasma membrane, the imaged receptors may partly be present at internal structures. After 20 min of continuous agonist addition, most receptors will be internalized. In recent years there have been countless papers from many research groups showing that the internal pool of receptors is functional and can signal in a compartment specific manner [57,58]. Future work should be conducted to show that the phenomenon observed occurs at the membrane and not an intracellular pool. This could be achieved in the presence of a dynamin inhibitor and/or by using cell impermeant agonists to confirm some of the key results. Fourth, live two-photon imaging was performed at room temperature, instead of at 37 °C, partly due to the weak signals of our fluorescent proteins and the known fluorescence enhancement of blue fluorescent proteins at low temperature [59]. Capturing the images at room temperature may have influenced the mean speed data and slow receptor internalization. Fifth, C-terminally tagging GPCRs can result in blocking PDZ ligands and masking binding partner motifs that are important to surface trafficking, internalization, and recycling. Future characterization of the rates of internalization of these MOP and NOP variations, as well as validation of the uniform expression of MOP and NOP on the cell surface under baseline conditions, would help address these concerns.

## 4. Materials and Methods

### 4.1. Molecular Cloning and Cell Culture

The full-length cDNA clones encoding the human MOP (Clone ID 30915262) and NOP (Clone ID 5164017) receptors purchased from Open Biosystems (Huntsville, AL, USA) were subcloned into mammalian expression vectors, pAmCyan1-N1 (CFP) and pZsYellow1-N1 (YFP) (Clontech, Mountain View, CA, USA), respectively. All sequences were verified by DNA sequence analysis. HEK 293 cells transiently transfected with human MOP-CFP and/or NOP-YFP using PolyJet Reagent (SignaGen Laboratories, Frederick, MD, USA) were grown in minimal essential medium (Thermo Fisher Scientific, Waltham, MA, USA) supplemented with 10% fetal bovine serum, 100 units/mL penicillin, and 100 μg/mL streptomycin. Cell cultures were maintained at 37 °C in a humidified 5% CO_2_ incubator [36].

### 4.2. FRET/SPT Measurements

HEK 293 cells were seeded in a µ-Dish (Ibidi GmbH, Gräfelfing, Germany) and transiently transfected with MOP-CFP and NOP-YFP, as described above. Cells were grown to near confluence and serum-starved overnight prior to FRET/SPT measurements. Morphine (1 μM, Toronto Research Chemicals, North York, ON, Canada) or buprenorphine (1 μM, Toronto Research Chemicals) was added to cells shortly before starting FRET/SPT measurements and present in the serum-free medium throughout the whole experiment. FRET/SPT on living cells was performed using a two-photon microscope (Bergamo, Thorlabs, Newton, NJ, USA) at room temperature. Notably, the only variable between the control and experimental groups was the difference between with and without drug treatment, instead of temperature. It is true that temperature may influence the mean speed data and slow receptor internalization. However, in the experiment, we kept everything identical (including temperature) and observed the phenomenon under drug interactions. Thus, we did not analyze the correlation between temperature and the phenomenon because it was not the goal of this work. The excitation source was provided by a tunable femtosecond laser system (Discovery, Coherent, Santa Clara, CA, USA). In quantitative cellular imaging, the laser was tuned to 810 nm and focused by a water-immersion objective lens (XLUMPLFLN 20X/1.0, Olympus, Japan) to excite cells expressing the CFP-YFP construct with a frame size of 150 × 150 μm^2^ on 1024 × 1024 pixels. To observe the long-term dynamics of receptor proteins in the downstream signaling pathways after drug treatment, a time course of 20 min with a 15 s time interval between each frame was used. Signals from CFP and YFP were collected by a condenser lens (CSC1003, 1.4 NA, Thorlabs, Newton, NJ, USA), separated by a dichroic beamsplitter (Di02-R514, Semrock, Rochester, NY, USA), sorted by respective filters (FF01-488/50 and FF01-545/55, Semrock, USA), and finally detected by two identical photomultiplier tubes (PMT2101/M, Thorlabs, USA) in the forward direction. Due to the spectral overlap between CFP and YFP resulting in considerable leak-through of the CFP signal into the YFP channel, the net FRET intensity was calibrated by imaging the cells expressing only the CFP or YFP constructs [60,61]. The equation used to remove the fluorescence cross-talk is given by:(1)$$\mathrm{Net}\;\mathrm{FRET}=\mathrm{YFP}\;\mathrm{siganl}-\left(a\times \mathrm{YFP}\;\mathrm{signal}\right)-\left(b\times \mathrm{CFP}\;\mathrm{signal}\right),$$
where *a* and *b* are the ratios of CFP/YFP signal to YFP/YFP signal in imaging cells expressing YFP construct only and CFP/YFP signal to CFP/CFP signal in imaging cells expressing CFP construct only, respectively. The numerator indicates the wavelength used for the specified fluorescent protein, while the denominator indicates the channel for collecting the fluorescence from the specified fluorescent protein. In this context, the wavelengths for exciting the CFP and YFP constructs only were 810 and 960 nm, respectively. After the intensity calibration with the *a* and *b* ratios, the image format was changed to 32 bits, and the image background was subtracted by the value measured in a random field outside of cells, which enabled us to obtain the ratio image of Net FRET/CFP that provided the floating result of the FRET ratio for each pixel. In the statistics, the regions of interest (ROIs) were selected within a cell presenting relatively uniform fluorescence on both channels, which was helpful in reducing the uncertainty and occurrence of unreasonable values (i.e., negative or larger than unity) in the FRET ratio. Finally, the results were summarized in a histogram to determine the value with maximum probability and standard deviation, i.e., the full width at half maximum (FWHM) of the histogram.

For the analysis of SPT, we focused on tracking the slowly moving MOP-NOP heterodimers mainly on membrane surfaces in the downstream signaling pathways (instead of the transient colocalization events in the early stimulation stage [13]), which were shown on both channels, thus enabling the retrieval of their moving trajectories with a 2D Gaussian fitting [62]. The measured trajectories were later used to calculate the mean speed and mean square displacements (MSD) of the heterodimers for determining the diffusion coefficient and mode of transport according to the slope in the log-log plot of MSD as a function of lag times. The theory of MSD and the relevant formula are described as follows [63,64]. In general, particles shown on an image contain X-Y position information, thus the 2D MSD can be calculated according to Equation (2).
(2)$$\mathrm{MSD}\left(t\right)=\mathrm{MSD}\left(n∆t\right)=\frac{1}{N-n}\overset{N-n}{\underset{i}{\sum }}[{\left(x_{i+n}-x_{i}\right)}^{2}+{\left(y_{i+n}-y_{i}\right)}^{2}],$$
where Δ*t* is the time interval between each frame, *N* is the total number of frames, *n* is the number of time intervals, and *x*_*i*_ and *y*_*i*_ are the positions for time point *i*. With Equation (2), the dependency of MSD on *t* is built to further determine the transport mode and diffusion coefficient of particles. For confined motion in our case, particles will exhibit anomalous diffusion, which is described as:(3)$$\mathrm{MSD}\left(t\right)=4Dt^{\alpha }+c,$$
where *D* is the microscopic diffusion coefficient when *α* < 1, *α* is the exponent used to confirm the confined motion of particles, and *c* is the offset term. Note that only the trajectory constituted by more than 50 successive frames was enrolled in the analysis.

To better understand the movement of MOP-NOP heterodimers upon opioid treatment, FRET/SPT measurements were performed repeatedly 6 times, in which 12 cells and 24 trajectories were analyzed for each condition and the subsequent data statistics by generating histograms based on the measured FRET ratio and mean speed in FRET and SPT experiments, respectively. The analyses of the FRET ratio and mean speed were performed using the built-in functions of ImageJ/Fiji software (National Institutes of Health, Bethesda, MD, USA), while a home-built MATLAB-based program (Mathworks, Natick, MA, USA) was used for the analysis of MSD as well as the diffusion coefficient.

### 4.3. Lipid Raft Labeling

Lipid rafts were labeled using the Vybrant^®^ Lipid Raft Labeling Kit (Thermo Fisher Scientific) with modifications. Briefly, HEK 293 cells stably expressing HA-tagged MOP and/or myc-tagged NOP [36] were seeded on sterile coverslips placed in 12-well culture dishes two days before the immunocytochemical assay. Cells were incubated with 1 μM morphine, buprenorphine, or vehicle (ddH_2_O) in the presence of 1 μg/mL (diluted in growth media) Alexa Fluor 555-labeled cholera toxin subunit B (CT-B) for 20 min at 4 °C. This CT-B conjugate bound to the pentasaccharide chain of plasma membrane ganglioside G_M1_ that selectively partitioned into lipid rafts. Subsequently, the cells were gently washed with chilled PBS and then incubated with rabbit anti-CT-B antibody for 30 min at 4 °C to crosslink the CT-B-labeled lipid rafts with the anti-CT-B antibody. After this incubation, the cells were gently washed with chilled PBS containing 1 mM MgCl_2_ and 0.1 mM CaCl_2_ (PBS^+^), fixed in freshly prepared 4% paraformaldehyde (in PBS^+^) for 30 min at 4 °C, washed again in PBS^+^, and permeabilized with 0.2% Triton X-100 in PBS^+^ for 15 min at room temperature. After washing with PBS^+^, the cells were blocked with 10% BSA in PBS^+^ for 30 min at room temperature, and immunostaining was performed by incubating the cells with a 1:100 dilution of HA-Tag (6E2) mouse monoclonal antibody (Cell Signaling Technology, Danvers, MA, USA) or c-Myc Tag chicken polyclonal antibody (Thermo Fisher Scientific) in 10% BSA in PBS^+^ overnight at 4 °C. The secondary antibody (1:500 dilution of Alexa Fluor 488 goat anti-mouse [Thermo Fisher Scientific] or Alexa Fluor 647 donkey anti-chicken IgY antibody [Millipore]) was applied for 2 h at room temperature. The cells were then washed with PBS^+^, mounted onto microscope slides with a ProLong^®^ Gold Antifade Kit (Thermo Fisher Scientific), and visualized using a Leica TCS SP8 X Confocal Spectral Microscope Imaging System with White Light Laser (Leica Microsystems, Wetzlar, Germany). Analysis of the degree of colocalization between lipid rafts and MOP or NOP immunoreactivity was performed using Imaris software (Bitplane, Zurich, Switzerland) [65]. Lipid rafts labeled by red fluorescence were defined as the ROI. Localization of the MOP and NOP receptors on the ROI was identified using the mono-color layers, i.e., the Alexa Fluor 488 and Alexa Fluor 647 layers, respectively. Colocalization coefficients were then calculated using the proprietary module of Imaris software. Statistical analysis was performed using Prism software (GraphPad Software, La Jolla, CA, USA).

### 4.4. Immunoblotting of Phosphorylated Erk1/2 (p44/p42 MAPK) Isoforms

HEK 293 cells harboring HA-tagged MOP and/or myc-tagged NOP [36] were grown to near confluence in 6-well plates, serum-starved for 16 h, and then incubated with various concentrations of morphine or buprenorphine for 10 min at 37 °C in a humidified 5% CO_2_ incubator. After being washed twice with ice-cold phosphate-buffered saline (PBS), cells were lysed in lysis buffer containing 50 mM Tris-HCl (pH 7.5), 150 mM NaCl, 1 mM MgCl_2_, 0.25% sodium deoxycholate, 0.2% Triton X-100, 10% glycerol, 2 mM NaF, 1 mM dithiothreitol, 0.5 mM phenylmethylsulfonyl fluoride, 0.2 mM sodium pyrophosphate, 1 mM sodium orthovanadate, and 1% protease inhibitor mixture (Sigma-Aldrich, St. Louis, MO, USA) for 15 min at 4 °C. Cell debris was precipitated by centrifugation at 14,000× *g* for 10 min at 4 °C, and the supernatant was used for the analysis. The protein concentration of the supernatant was determined using the Pierce^TM^ BCA assay (Thermo Fisher Scientific, Waltham, MA, USA) with bovine serum albumin as the standard. Proteins were resolved using 10% SDS polyacrylamide gels in duplicate and then transferred to polyvinylidene difluoride membranes. Membranes were incubated with anti-phospho-p44/p42 MAPK (Erk1/2) (Thr202/Tyr204) rabbit polyclonal or anti-p44/42 MAPK (Erk1/2) (137F5) rabbit monoclonal antibody (Cell Signaling Technology, Danvers, MA, USA) overnight at 4 °C. After being washed, the membrane was incubated with Amersham donkey anti-rabbit horseradish peroxidase-linked secondary antibody (GE Healthcare, Little Chalfont, Buckinghamshire, UK). Subsequently, immunoreactive proteins on the membrane were revealed by enhanced chemiluminescence (SuperSignal^TM^ West Pico chemiluminescent substrate; Thermo Fisher Scientific). Images were captured using Image Lab^TM^ Software (Bio-Rad, Hercules, CA, USA). The intensities of the bands on the immunoblots were quantified using Image Lab^TM^ software (Bio-Rad), and then a four-parameter dose–response curve was applied to determine the pEC50 of the drug using Prism 6 software (GraphPad Software, San Diego, CA, USA).

## 5. Conclusions

Our study demonstrates how opioids modulate the cellular localization of MOP and NOP receptors (membrane-delimited) as well as Erk1/2 (a cytosolic effector) in a cell model coexpressing human MOP and NOP receptors, revealing cross-talk between coexpressed MOP and NOP receptors in cellular signaling after treatment with morphine or buprenorphine. Dimerization and colocalization of MOP and NOP on lipid rafts was enhanced by morphine but not buprenorphine. Erk1/2 activation by morphine via MOP was not hampered by coexpressed NOP, while buprenorphine appeared to act mainly on NOP to activate Erk1/2.

## Figures and Tables

**Figure 1 ijms-24-01048-f001:**
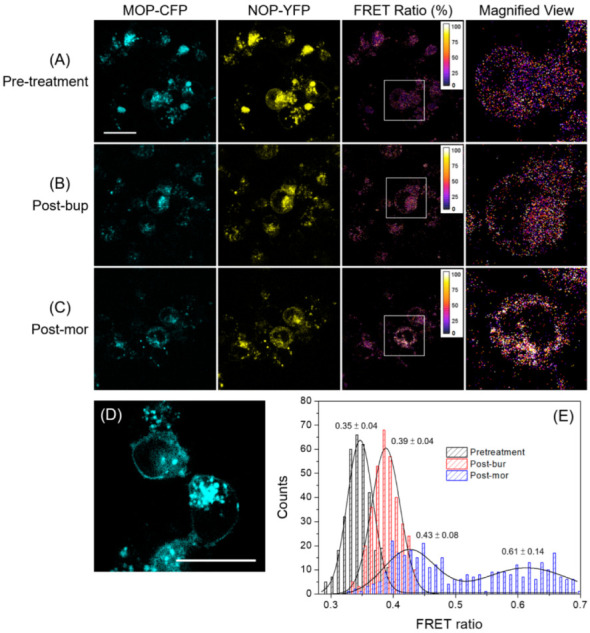
HEK 293 cells transiently transfected with MOP-CFP and NOP-YFP were serum-starved overnight; vehicle (ddH_2_O), morphine (1 μM), or buprenorphine (1 μM) was added to cells immediately before FRET/SPT measurements. (**A**–**C**) present the results obtained under the conditions of pre-treatment, post-bup, and post-mor, respectively. The images of MOP-CPF (shown in cyan), NOP-YFP (shown in yellow), FRET ratio (%), and a magnified view from a white box in the respective FRET ratio image are shown for each condition from left to right. (**D**) is the control image taken from the MOP-CFP-channel at the pretreatment condition, which is used to confirm whether the receptors are properly expressed on the cell surface. (**E**) The histograms of the FRET ratio obtained from each condition were fitted to Gaussian distributions, indicating the peak value with a standard deviation. Scale bar = 20 μm. The bin size of the histogram = 0.01. Twelve cells were used for statistical analysis in each condition.

**Figure 2 ijms-24-01048-f002:**
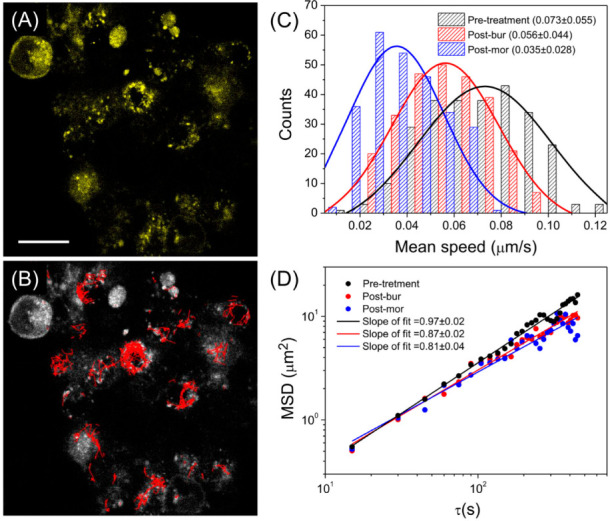
(**A**,**B**) are 2D images taken from the NOP-YFP channel treated with morphine, with (**B**) showing the trajectories of mobilized heterodimers. Scale bar = 20 µm. (**C**) is the histogram of mean speed for the three conditions, which were fitted to Gaussian distributions showing the mean speed with maximum probability and a standard deviation. (**D**) presents the extraction of the exponent and diffusion coefficient in the three conditions from linear fitting to the log-log plot of MSD with respect to lag times. To provide a clear interpretation of MSD data, standard deviations are not shown. Twenty-four trajectories were used for statistics in each condition.

**Figure 3 ijms-24-01048-f003:**
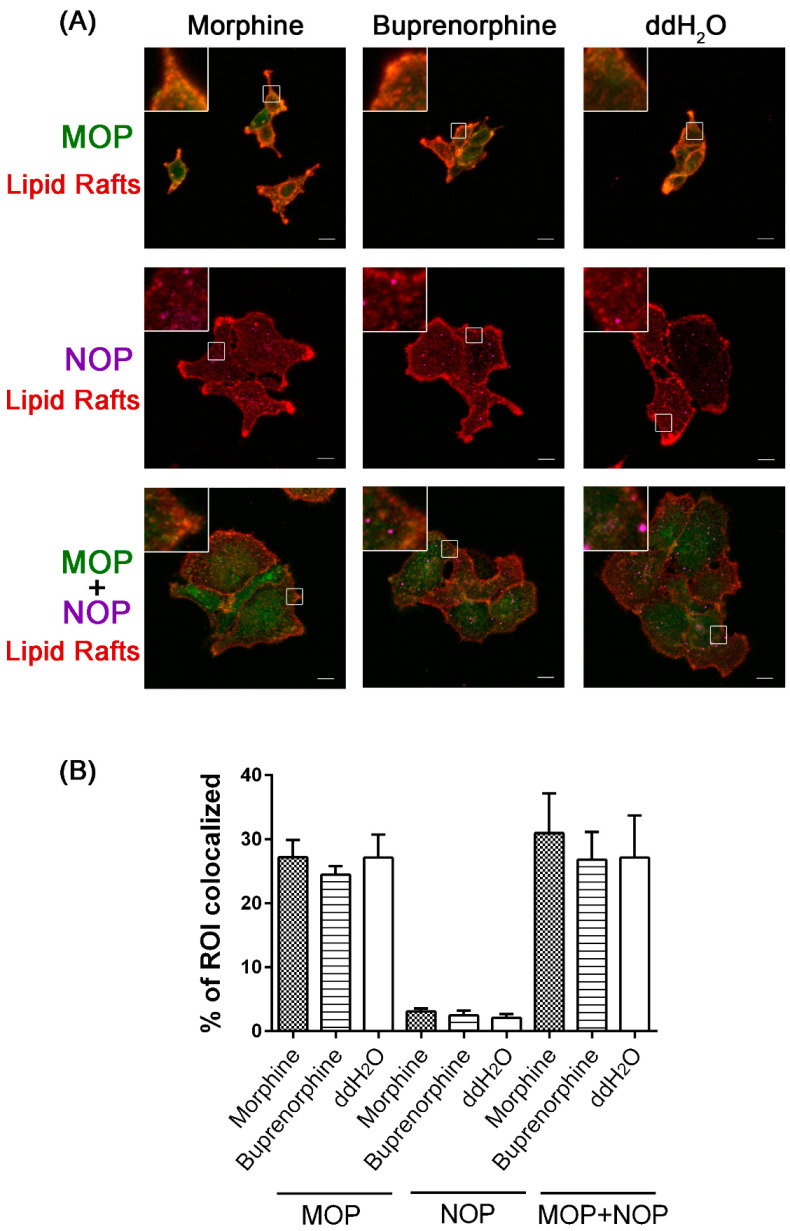
Representative confocal images (**A**) and colocalization rates of MOP and NOP on lipid rafts (red) (**B**) from HEK 293 cells stably expressing HA-tagged MOP (green) and/or myc-tagged NOP (magenta) after exposure to morphine (1 μM) or buprenorphine (1 μM) for 20 min at 4 °C. The inset panels display higher magnification images highlighting the colocalization. Scale bars are equal to 10 μm. Each bar represents the mean ± SE value of the percentage of ROI colocalization derived from five independent experiments with 4–6 individual cell images in each group.

**Figure 4 ijms-24-01048-f004:**
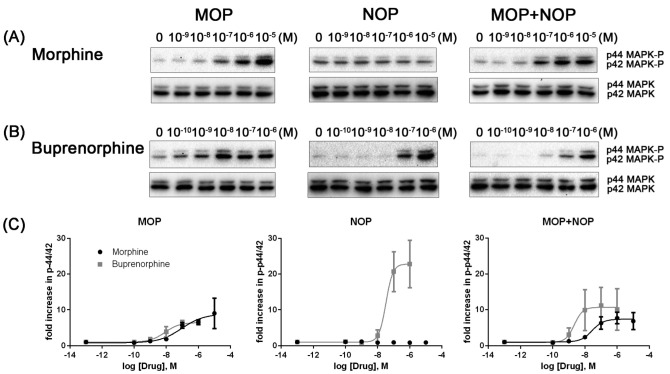
Representative immunoblots of Erk1/2 (p44/p42 MAPK) phosphorylation after stimulation with (**A**) morphine and (**B**) buprenorphine in cells stably expressing MOP, NOP, and MOP+NOP receptors. Immunoblotting using anti-phospho-p44/p42 MAPK (Erk1/2) rabbit polyclonal and anti-p44/42 MAPK (Erk1/2) rabbit monoclonal antibodies was performed to visualize opioid-induced phosphorylation of Erk1/2 (p44 MAPK-P and p42 MAPK-P) and total Erk1/2 (p44 MAPK and p42 MAPK), respectively. (**C**) Dose–response curves of the immunoblots. The density of phospho-p44/p42 MAPK versus the density of p44/42 MAPK in control cells (with vehicle treatment) was considered to be 1. All the experiments were repeated no less than four times, and the values represent the average ± SE.

## Data Availability

The data that support the findings of this study are available from the corresponding author upon reasonable request.

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
