# Peer review of "Opioid-Modulated Receptor Localization and Erk1/2 Phosphorylation in Cells Coexpressing μ-Opioid and Nociceptin Receptors"

_ijms, 2023, doi:10.3390/ijms24021048_

Round 1

Reviewer 1 Report

The study presented for review aimed to investigate the differences in the dynamic of MOP-NOP receptor dimer formation by morphine and buprenorfine as basis for their differential liability in producing opioid-mediated side effects. Those include analgesic tolerance and withdrawal, mediated by AC superactivation and/or enhanced beta-arrestin recruitment and subsequent Erk1/2 activation. The interesting concept that AC superactivation is dependent on the abundance of MOP receptors within the lipid rafts of the lipid membrane was also put forward. As well as that MOP-NOP dimer formation might facilitate Erk1/2 activation that may accounts for e.g tolerance development. This work presents an interesting concept of a clinically relevant MOP and NOP interaction, however few major issues need to be resolved:

  1. In the Abstract section the Authors state that morphine stimulated Erk1/2 phosphorylation with equal potency in both MOP only and MOP+NOP expressing cells. Does that mean that enhancing MOP and NOP colocalization is of little importance for Erk1/2 activation? Wouldn’t it seem plausible that AC hyperactivation could also be a reliable molecular underpinning for morphine-induced central side effects? Especially when the Authors state that the MOP-NOP heterodimer mediates MOP inactivation. It is possible that this happens because of a putative G-protein switch and AC hyperactivation. Has AC activity or cAMP levels been studied? If so, it’s important to include those results in the paper.
  2. The morphine-induced colocalization of MOP and NOP shown in Fig. 3B seems statistically insignificant as the error bars overlap. There is little information about the statistical analysis performed and also no indication of statistical differences has been provided in Fig. 3B. There seems to be a minute tendency for stronger colocalization in MOP+NOP cells upon morphine treatment, however the same tendency with a similar magnitude is also visible for MOP expressing cells.
  3. Quantification of immunoblot results in Fig. 4 is lacking, which makes interpretation of results difficult. It makes it very hard to discern which comparisons are significant if no bar graphs are provided.
  4. Does the heterodimerization process differ in cells that don't overexpress MOP and NOP receptors? In functional assays, drug efficacy can markedly differ between cell systems with overexpressed receptors compared with native systems. Especially for partial agonists. Morphine is a partial agonist in native membrane systems (e.g the hypothalamus) and a full agonist in cells overexpression mu opioid receptors. Maybe a similar rule applies to beta-arrestin recruitment and Erk1/2 activation?
  5. It would be informative whether the effects observed are reversed with selective antagonists and what is the effect of selective MOP and NOP agonists that produce receptor internalization. As the side effects of morphine might stem from the lack of receptor internalization and recycling.
  6. It’s my own personal opinion, the discussion should emphasize the clinical perspective of the results obtained. What is the importance of the delay in MOP+NOP motion?

Minor issues:

  1. Lines 56-57 – be more specific and provide information on what profound clinical consequences could be observed following MOP and NOP coupling.
  2. Line 280 – Avoid ambigious terms like “morphine and buprenorphine differentially regulated intracellular AC activity in MOP+NOP-expressing cells”. Please be more specific and indicate the direction of this effect.
  3. Line 278 – what does it mean that morphine “is weak”? That it weakly recruits beta-arrestin2?

Reviewer 2 Report

Opioid-Modulated Receptor Localization and Erk1/2 Phosphor-2ylation in Cells Coexpressing μ-Opioid and Nociceptin Receptors

Summary:

         The author’s goal is to show that exogenous opioid agonist addition can modulate both the surface localization into lipid rafts and co-localization of MOP-NOP receptors as well as influence their downstream signaling. For this study, they employ 2-photon microscopy and FRET to examine the MOP and NOP colocalization at the membrane. While the general observations are supported by the data, additional work is needed verify the true heterodimerization of MOP-NOP at the cell surface. The manuscript is well written and organized and was very pleasant to read, but some of the Figures need improved. Overall, this article provides an interesting use case for 2-photon microscopy in the field of receptor trafficking and begins to scratch the surface of the MOP-NOP heterodimerization. After addressing the following concerns, this manuscript could be considered for publication. 

Major Concerns: 

1.     The authors state that the main reasons to use 2-photon microscopy is to avoid light loss via the confocal pin hole and to minimize photobleaching. While 2-photon microscopy avoids photobleaching above and below the focal plane, it can result in more photobleaching during long-term live imaging experiments. Was the system used a resonant scanning confocal or galvo mirror system? This would seem to have a larger impact on live imaging. Further, the optical sectioning of 2-photon imaging is similar to confocal and so light collection is also similar. Lastly, a power usage of 25 mW is given and explained as being low power. For most visible current confocal laser lines the max power is between 25-50 mW. This means that the power used here is equivalent to around 50% power in a normal confocal system which is actually high wattage. The first paragraph in section 2.1 should be reviewed and rewritten to address these concerns. 

2.     The authors state on page 3 that “However, the function of the receptors on the cell membrane is to receive drugs and to produce downstream signals. Thus, the phenomena observed by FRET and later by SPT are mainly from membrane surfaces.” I do not think this is valid. After 20 min of continuous agonist addition most receptors will be internalized. In recent years there have been countless papers from many groups showing that the internal pool of receptors are functional and can signal in a compartment specific manner (ex. doi: 10.7554/elife.67478, doi: 10.1016/j.tips.2017.11.009). There should be more work done to show that the phenomenon observed is occurring at the membrane and not an intracellular pool. This could be done in the presence of a dynamin inhibitor and/or by using cell impermeant agonists to confirm some of the key results.   

3.     C-terminally tagging GPCRs can result in blocking of the PDZ ligands and masking of binding partners motifs important to surface trafficking, internalization, and recycling. Have the authors characterized the rates of internalization of these MOP and NOP variations? Additionally, can the authors show control conditions where these receptors are nicely and uniformly expressed on the cell surface under baseline conditions? 

4.     In Figure 1, the cells look very round and are not spread out on the coverslip like HEK cells normally look. Have the authors confirmed that the cells are mycoplasma negative or is this a result of the plasmid transfection or imaging conditions? 

5.     For all imaging data, it would be helpful to show a high magnification inset panel with the figures to more clearly be able to appreciate the FRET and co-localization results.  

6.     As currently organize Figure 3 is difficult to interpret. The images should be better labeled as to was is red and what is green. I see the red channel is the Vybrant kit label but this should be clear from the figure itself and not rely on the legend.  Higher magnification images are needed to see the co-localization. Please include statistics on panel B. Some of the detailed methods are not appropriate for the figure legend. 

7.     For Figure 4, can the authors provide the densitometry in either a table or graph format? This would be for comparison and statistics determination. 

8.     The authors mention the activation of Src by MOP in lipid rafts and suggest that it would be something they should look into in the future. It would be a nice addition to this manuscript if immunoblotting for Src and pSrc were added to Figure 4. This would strengthen the lipid raft conclusions. 

Minor Concerns: 

1.     Was the live 2-photon imaging performed at 37ºC? If not, this could influence the mean speed data and slow receptor internalization. Please expand on these details and justify why room temperature imaging was appropriate if used. 

2.     For the histograms in Figure 1, it would be helpful to have all the graphs on the same X-axis scale. 

3.     In Figure 2B, the particle tracking traces would be more easily seen if they were a color other than yellow since panel A is also yellow.  

4.     Remove the words “first to demonstrate” from the article. This does not add additional novelty. 

Round 2

Reviewer 1 Report

I am satisfied with the responses to the review

Author Response

We really appreciate your positive comment!

Reviewer 2 Report

Thank you for the added changes. The figures and paper are much clearer. There is still one major concern and a few minor issues. 

From the changes made it still is not clear why the authors feel that 2-Photon microscopy is better for live cell imaging than visible confocal or resonant scanning or spinning disk confocal? Further, it appears that some of the data might be either hindered by the low signal of MP and need for higher laser power that might be causing the cell rounding and potential artifacts. I would recommend removing all claims about the 2P imaging being better since the data does not seem to support this claim and even the visible confocal data in Figure 3 looks better than the FRET images. 

The authors do mention that problems with the cells might be due to transfection issues. Due to this concern, it is necessary to show at least as a control in one of the figures that the receptors are properly expressed on the cell surface via an imaging method of your choosing. All of the current data relies on this assumption.   

Minor Concerns: 

Figure 3A, NOP is not easily seen in the colocalized images. Can you add the individual channel images in grayscale to a supplemental Figure? 

Figure 3 B. Mention in the text that these changes are not significant since there is no reported statistics as mentioned in the review response. 

Figure 4 Panel C make the Y-axis the same scale to compare folds change between treatments. 

Round 3

Reviewer 2 Report

Thank you for addressing the concerns. The manuscript is much improved.